

# Anti-predator defences of a bombardier beetle: is bombing essential for successful escape from frogs?

Shinji Sugiura

Graduate School of Agricultural Science, Kobe University, Kobe, Japan

## ABSTRACT

Some animals, such as the bombardier beetles (Coleoptera: Carabidae: Brachinini), have evolved chemical defences against predators. When attacked, bombardier beetles can discharge noxious chemicals at temperatures of approximately 100 °C from the tip of their abdomens, "bombing" their attackers. Although many studies to date have investigated how bombardier beetles discharge defensive chemicals against predators, relatively little research has examined how predators modify their attacks on bombardier beetles to avoid being bombed. In this study, I observed the black-spotted pond frog *Pelophylax nigromaculatus* (Anura: Ranidae) attacking the bombardier beetle *Pheropsophus jessoensis* under laboratory conditions. In Japan, *Pe. nigromaculatus* is a generalist predator in grasslands where the bombardier beetle frequently occurs. Almost all the frogs (92.9%) observed rejected live bombardier beetles; 67.9% stopped their attacks once their tongues touched the beetles, and 25.0% spat out the beetles immediately after taking the beetles into their mouths. No beetle bombed a frog before being taken into a frog's mouth. All beetles taken into mouths bombed the frogs. Only 7.1% of the frogs swallowed live bombardier beetles after being bombed in the mouth. When dead beetles were provided instead, 85.7% of the frogs rejected the dead beetles, 71.4% stopped their attacks after their tongues touched the beetles, and 14.3% spat out the beetles. Only 14.3% of the frogs swallowed the dead beetles. The results suggest that the frogs tended to stop their predatory attack before receiving a bombing response from the beetles. Therefore, bombing was not essential for the beetles to successfully defend against the frogs. Using its tongue, *Pe. nigromaculatus* may be able to rapidly detect a deterrent chemical or physical characteristics of its potential prey *Ph. jessoensis* and thus avoid injury by stopping its predatory attack before the beetle bombs it.

## INTRODUCTION

Physical and chemical defences have evolved in many organisms to protect against natural enemies (*Edmunds, 1974*; *Eisner, Eisner & Siegler, 2005*). For example, some plant and animal species have developed physical deterrents such as thorns and spines (*Edmunds, 1974*; *Cooper & Owen-Smith, 1986*; *Eisner, 2003*; *Sugiura & Yamazaki, 2014*; *Sugiura, 2016*; *Ito, Taniguchi & Billen, 2016*), while other species produce defensive chemicals, including toxic substances, to prevent themselves from being eaten (*Eisner, 2003*; *Eisner, Eisner & Siegler, 2005*; *Derby, 2007*; *Mithöfer & Boland, 2012*). Organisms whose defence

Corresponding author
Shinji Sugiura,
ssugiura@people.kobe-u.ac.jp,
sugiura.shinji@gmail.com

mechanisms can cause severe injury to their natural enemies have also evolved warning signals, such as conspicuous body colouration or particular sounds (*Lev-Yadun, 2001*; *Ruxton, Sherratt & Speed, 2004*; *Inbar & Lev-Yadun, 2005*; *Bonacci et al., 2008*; *Lev-Yadun, 2009*; *Bura, Kawahara & Yack, 2016*; *Sugiura & Takanashi, 2018*). In response, predators have evolved specific abilities to avoid such well-defended prey by recognising warning colouration or detecting chemical signals (*Edmunds, 1974*; *Endler, 1991*; *Ruxton, Sherratt & Speed, 2004*; *Skelhorn & Rowe, 2006*; *Williams et al., 2010*).

Adult bombardier beetles (Coleoptera: Carabidae: Brachinini) bomb, i.e., discharge noxious chemicals from the tip of their abdomens at temperatures of approximately 100 °C, when they are disturbed or attacked (*Aneshansley et al., 1969*; *Dean, 1979*; *Eisner, 2003*; *Eisner, Eisner & Siegler, 2005*; *Arndt et al., 2015*). Such ejection of hot chemicals is only known in the coleopteran family Carabidae. Previous studies have investigated how bombardier beetles successfully defend against predators (*Eisner, 1958*; *Eisner & Meinwald, 1966*; *Eisner & Dean, 1976*; *Dean, 1980a*; *Eisner, Eisner & Aneshansley, 2005*; *Eisner et al., 2006*). Bombardier beetles can aim their abdominal discharge in virtually any direction, spraying various parts of their own bodies (e.g., legs and dorsal surface) with the toxic chemicals (*Eisner & Aneshansley, 1999*). *Dean (1980b)* reported that predators displayed intense responses to the unheated chemical discharges of bombardier beetles in experiments. This suggests that the cooled chemicals coating the beetles' body surfaces function as the primary defence against predators. Successful defence mediated by chemicals on the body surfaces of beetles may reduce the costs of spraying (bombing). Further research is needed to clarify the relative importance of chemical toxicity and heat for overall successful anti-predatory defence.

Frogs and toads are important predators of carabid beetles (*Larochelle, 1974a*; *Larochelle, 1974b*). However, bombardier beetles have rarely been found in the gut contents and faeces of frogs and toads (*Larochelle, 1974a*; *Larochelle, 1974b*; *Sarashina, Yoshihisa & Yoshida, 2011*; except *Mori, 2008*), suggesting that bombing prevents toads and frogs from swallowing and ingesting beetles (*Eisner & Meinwald, 1966*; *Dean, 1980a*; *Eisner, 2003*; *Sugiura & Sato, 2018*). Still, only a few studies have investigated the factors that cause anuran predators to stop preying on bombardier beetles (*Dean, 1980b*). Elucidating these ecological factors would contribute to a better understanding of the evolution of anti-predatory defences in insects.

This study aims to investigate the responses of the black-spotted pond frog *Pelophylax nigromaculatus* (Hallowell) (Anura: Ranidae) to the defensive behaviour of the bombardier beetle *Pheropsophus jessoensis* (Morawitz). *Pheropsophus jessoensis* is a bombardier beetle found in East Asia (*Ueno, Kurosawa & Sato, 1985*; *Jung et al., 2012*); the beetle is a common inhabitant of farmlands, grasslands, and forest edges in Japan (*Habu & Sadanaga, 1965*; *Ueno, Kurosawa & Sato, 1985*; *Yahiro et al., 1992*; *Ishitani & Yano, 1994*; *Fujisawa, Lee & Ishii, 2012*; *Ohwaki, Kaneko & Ikeda, 2015*; *Sugiura & Sato, 2018*). Adult *Ph. jessoensis* eject toxic chemicals (1,4-benzoquinone and 2-methyl-1,4-benzoquinone) at a temperature of approximately 100 °C from their rear ends in response to predator attacks (Video S1; *Kanehisa & Murase, 1977*; *Kanehisa, 1996*). *Pelophylax nigromaculatus* is a true frog inhabiting wetlands and farmlands of East Asia (*Liu et al., 2010*; *Tsuji et al., 2011*; *Komaki et*

*al., 2015*), being one of the most abundant frog species of traditional agricultural landscapes including farmlands, grasslands, and forest edges (*Hirai, 2002*; *Honma, Oku & Nishida, 2006*; *Tsuji et al., 2011*; *Matsuhashi & Okuyama, 2015*). Using its tongue, *Pe. nigromaculatus* readily catches and swallows smaller prey (Video S2; *Honma, Oku & Nishida, 2006*). *Pelophylax nigromaculatus* is a generalist predator that has been reported to prey on carabid beetles (*Maeda & Matsui, 1999*; *Hirai & Matsui, 1999*; *Sano & Shinohara, 2012*; *Sarashina, Yoshihisa & Yoshida, 2011*). As *Ph. jessoensis* and *Pe. nigromaculatus* co-occur in the same grassland habitats, this frog species is a potential predator of adult *Ph. jessoensis*. In early June 2016, I offered an adult *Ph. jessoensis* to an adult *Pe. nigromaculatus* under laboratory conditions. The frog attacked the beetle, but stopped the attack immediately after its tongue touched the beetle. No bombing sounds were heard, suggesting that the frog ceased its attack before the beetle bombed. Therefore, I hypothesised that bombing is not essential when *Ph. jessoensis* seeks to avoid being swallowed by *Pe. nigromaculatus*. To test this hypothesis, I observed *Pe. nigromaculatus* attacking *Ph. jessoensis* under laboratory conditions using a digital video camera. Acceptance or rejection of prey was carefully investigated using slow-motion videos. Furthermore, both dead and live beetles were used to test whether bombing is essential for successful defence against predatory attacks by *Pe. nigromaculatus*. Finally, I discuss the importance of primary and secondary defences in terms of overall anti-predation defence.

## MATERIALS AND METHODS

### Sampling

Approximately 100 adult *Ph. jessoensis* were collected from grasslands and forest edges in Kato-shi (34°54′N, 135°02′E, 120 m above sea level), Hyogo Prefecture, central Japan, from May to August in 2016, 2017, and 2018 (cf. *Sugiura & Sato, 2018*). Body weight was measured to the nearest 0.1 mg using an electronic balance (PA64JP, Ohaus, Tokyo, Japan). Study individuals were maintained separately in plastic cases (diameter: 85 mm; height: 25 mm) with wet tissue paper in the laboratory at 25 °C. Dead larvae of *Spodoptera litura* (Fabricius) (Lepidoptera: Noctuidae) were provided as food (*Sugiura & Sato, 2018*). Beetles were not used repeatedly in different feeding experiments. All experiments were conducted 22.6 ± 4.0 (means ± standard errors; range: 4–69) days after the beetles were collected.

Approximately 100 individuals of *Pe. nigromaculatus* were collected from wetlands and forest edges in Takarazuka-shi (34°53′N, 135°17′E, 230 m above sea level), Sanda-shi (34°57′N, 135°11′E, 180 m above sea level), and Sayo-cho (35°02′N, 134°20′E, 180 m above sea level), Hyogo Prefecture, central Japan, from May to August in 2016, 2017, and 2018. The distances between these sites and the sampling site of *Ph. jessoensis* ranged from 15.6 to 65.4 km. Although *Pe. nigromaculatus* has recently been classified as near threatened (NT) in the Japanese Red Data List (Ministry of the Environment of Japan, 2017), this species was abundant at the collection sites. Both juveniles and adults were collected. Body weight was measured to the closest 0.01 g using an electronic balance (EK-120A, A&D, Tokyo). Small and large frogs were maintained separately in small (120 × 85 × 130 mm, length ×

width × height) and large plastic cages (120 × 185 × 130 mm, length × width × height), respectively, in the laboratory at 25 °C. Live larvae of *S. litura*, *Tenebrio molitor* Linnaeus (Coleoptera: Tenebrionidae), and *Zophobas atratus* Fabricius (Coleoptera: Tenebrionidae) were provided as food. Frogs were starved for 24 h before the feeding experiments to standardise their hunger level (cf. *Honma, Oku & Nishida, 2006*). As with the beetles, individual frogs were not used repeatedly. The experiments were conducted 18.6 ± 2.5 (means ± standard errors; range: 4–66) days after the frogs were collected. The frogs were released after the experiments were completed.

## Feeding experiments

Feeding experiments were all conducted at 25 °C. To start, a frog was placed in a transparent plastic container (120 × 85 × 130 mm, length × width × height). Then, a transparent glass petri dish (45 mm in diameter, 15 mm in height) containing a live bombardier beetle was placed outside the plastic container where the frog could see it. Frogs that did not try to attack the beetle (34.1%) were not used for the feeding experiments. However, frogs that ignored *Ph. jessoensis* did not respond to other prey (i.e., *T. molitor* larvae). If a frog displayed attacking behaviour (i.e., opening the mouth and shooting out the tongue to capture prey; Video S2), a live beetle was then placed in the container with the frog. The resulting behaviours were recorded on video using a digital camera (iPhone 6 plus, Apple) at 240 frames per second. If the frog did not swallow the beetle, palatable prey (a *T. molitor* larva) was offered to the frog several minutes after beetle rejection to determine whether the frog was hungry. If a frog swallowed the bombardier beetle, I observed whether it vomited the beetle within 330 min of swallowing it (cf. *Sugiura & Sato, 2018*). Vomited beetles were checked to see whether they were still alive. Frogs that did not vomit after swallowing were considered to have digested the beetle. Frog faeces were examined after the experiment to confirm whether the beetles were digested. In total, 28 frogs and 28 live bombardier beetles were used in the experiments. The means ± standard errors of the frog and live beetle body weights were 10.23 ± 1.39 g ($n = 28$) and 213.0 ± 10.0 mg ($n = 28$), respectively.

A second set of frogs were presented with dead adult beetles to test whether the bombing response is essential for deterring a predatory attack. *Pelophylax nigromaculatus* usually does not attack motionless prey. However, in a pilot test, an individual of *Pe. nigromaculatus* attacked and ingested a dead caterpillar (*S. litura*) when forceps were used to move the caterpillar within the frog's field of view. For this experiment, the bombardier beetles were killed in a freezer at −15 °C. First, a dead beetle was placed in the plastic container (120 × 85 × 130 mm, length × width × height) within the frog's field of view. If the frog did not initially respond to the beetle, forceps were used to move the dead beetle within the frog's field of view again. Frogs that did not attack the dead beetles (29.2%) were not used in these experiments; frogs that ignored *Ph. jessoensis* did not respond to other prey (i.e., *T. molitor* larvae). The predatory behaviours of the frogs were recorded using the same digital video camera. Frogs that did not swallow dead beetles were offered *T. molitor* larvae several minutes after beetle rejection to check whether they were hungry. Twenty-eight frogs and 28 dead beetles were used in this experiment. The means ± standard errors of the frog and dead beetle body weights were 8.65 ± 1.25 g ($n = 28$) and 214.8 ± 7.6 mg

($n = 28$), respectively. The mean body weight of frogs that attacked dead beetles did not differ significantly from the mean body weight of frogs that attacked live beetles ($t$-test, $t = 0.84$, $P = 0.40$). The mean body weight of dead beetles did not differ significantly from the mean body weight of live beetles ($t$-test, $t = -0.14$, $P = 0.89$)

Videos of frogs responding to live and dead beetles were played back using QuickTime Player version 10.4 (Apple, Inc.). Frog responses to the bombardier beetles were grouped into four categories (cf. *Ito, Taniguchi & Billen, 2016*; *Matsubara & Sugiura, 2017*; *Sugiura & Sato, 2018*): (1) frogs that touched the beetles with their tongues but did not take the beetles into their mouths; (2) frogs that spat out the beetles after taking them into the mouth; (3) frogs that swallowed beetles but vomited them later; and (4) frogs that swallowed and digested the beetles. I also assessed whether frogs that rejected beetles (1–2) resumed their attacks within 10 s.

All experiments were performed in accordance with the Kobe University Animal Experimentation Regulations (Kobe University Animal Care and Use Committee, 27–01, 30–01). No frogs were seriously injured or killed during the feeding experiments. My study also complies with the current laws of Japan.

### Data analysis

Generalised linear models (GLMs) featuring binomial error distributions and logit links (i.e., logistic regressions) were used to identify factors that contributed to frogs' successful swallowing and digestion of the bombardier beetles. The success or failure (1/0) of frogs' swallowing and digesting beetles was used as the response variable. Frog weight, beetle weight, the frog weight × beetle weight interaction, and beetle condition (live or dead) were treated as fixed factors. When the residual deviance was larger (overdispersion) or smaller (underdispersion) than the residual degrees of freedom, a quasi-binomial error distribution was used rather than a binomial error distribution. Furthermore, fixed factors were subjected to likelihood ratio testing when marginal significance was evident. Thus, the significance of models with and without the factors of interest were compared using the GLMs. All analyses were performed using R version 3.3.2 (*R Development Core Team, 2016*).

## RESULTS

In the experiment using live adult bombardier beetles ($n = 28$), 26 frogs (92.9%) rejected the beetles without swallowing them (Fig. 1); 19 frogs (67.9%) stopped attacking the beetles immediately after touching the beetles with their tongues (Fig. 2; Video S3), and seven frogs (25.0%) spat out the beetles after taking the beetles into their mouths (Fig. 3; Video S4). No beetle bombed a frog before being taken into the frog's mouth. All beetles that were taken into frog mouths bombed (Video S4). Only two frogs (7.1%) were observed to swallow the bombardier beetles (Table 1) after being bombed in the mouth; one of the frogs successfully digested the beetle, but the other frog vomited the beetle 18 min after swallowing it (Table 1). The vomited beetle was still alive. Of the frogs that took the beetles into their mouths, 88.9% ($n = 8/9$) initially stopped attacking the beetles when their tongues first touched the beetles, but resumed their predatory attack soon thereafter

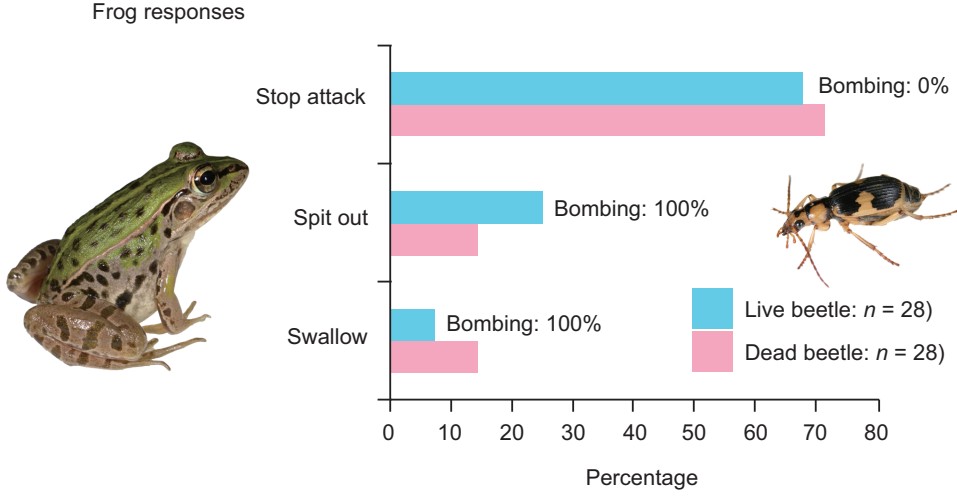

**Figure 1** **Behavioural responses of the black-spotted pond frog *Pelophylax nigromaculatus* to live and dead adult individuals of the bombardier beetle *Pheropsophus jessoensis*.** 'Stop attack': the frogs stopped their attacks after their tongues touched the beetles. 'Spit out': the frogs spat out the beetles immediately after taking the beetles into their mouths. 'Swallow': the frogs successfully swallowed the beetles. 'Bombing': the beetles could be heard bombing. Photo credit: Shinji Sugiura.

(Fig. 3; Video S4). Frogs that did not swallow beetles consumed other prey (i.e., *T. molitor* larvae) soon thereafter.

When dead beetles were used ($n = 28$), 24 frogs (85.7%) rejected the dead beetles without swallowing them (Fig. 1); 20 frogs (71.4%) stopped attacking the beetles after their tongues touched the dead beetles (Video S5), and four frogs (14.3%) spat out the beetles after taking the beetles into their mouths (Fig. 1). Only four frogs (14.3%) swallowed the dead beetles. Similar to the experiment using live beetles, 87.5% of the frogs that took beetles into their mouths ($n = 7/8$) were initially deterred when their tongues first touched the beetles, but continued with their predatory behaviour soon afterwards. The frogs that did not swallow beetles ate other prey (i.e., *T. molitor* larvae) soon thereafter.

The proportion of dead beetles swallowed by frogs (14.3%) was higher than that of live beetles (7.1%). However, the GLM results indicated that the frog swallowing rates of live and dead beetles did not significantly differ (Table 1). Whether beetles were swallowed or not was associated with beetle size, but not frog size (Table 1). Beetles were more likely to be swallowed as beetle size decreased (Fig. 4A). The interaction of frog and beetle weight was not significant (Fig. 4A).

The proportion of dead beetles digested by frogs (14.3%) was higher than the proportion of live beetles swallowed (3.6%). The GLMs indicated that the difference between the digestion rates of live and dead beetles was significant (Table 2). Beetle size affected the digestion rate (Table 2, Fig. 4B).

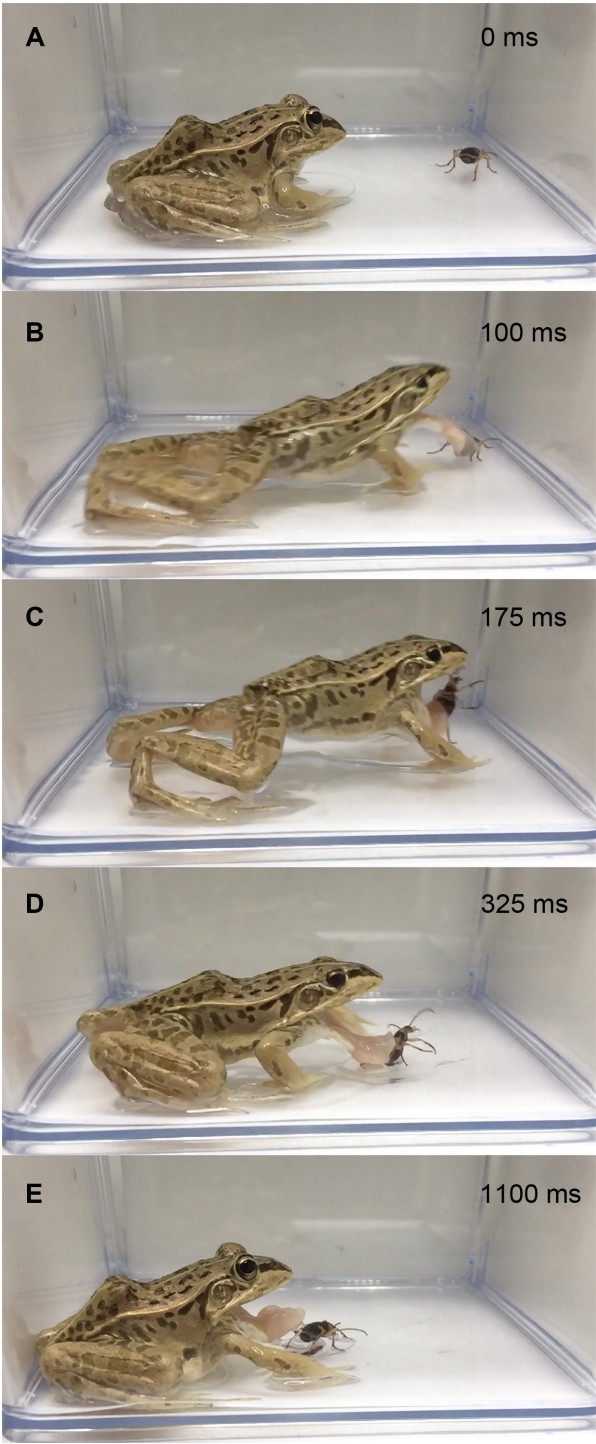

**Figure 2** Temporal sequence of the frog *Pelophylax nigromaculatus* rejecting a live adult *Pheropsophus jessoensis* without taking the beetle into its mouth. (A) 0 ms. (B) 100 ms. (C) 175 ms. (D) 325 ms. (E) 1,100 ms. The frog stopped the attack immediately after its tongue touched the beetle. No bombing sounds were heard (see Video S3). Credit: Shinji Sugiura.

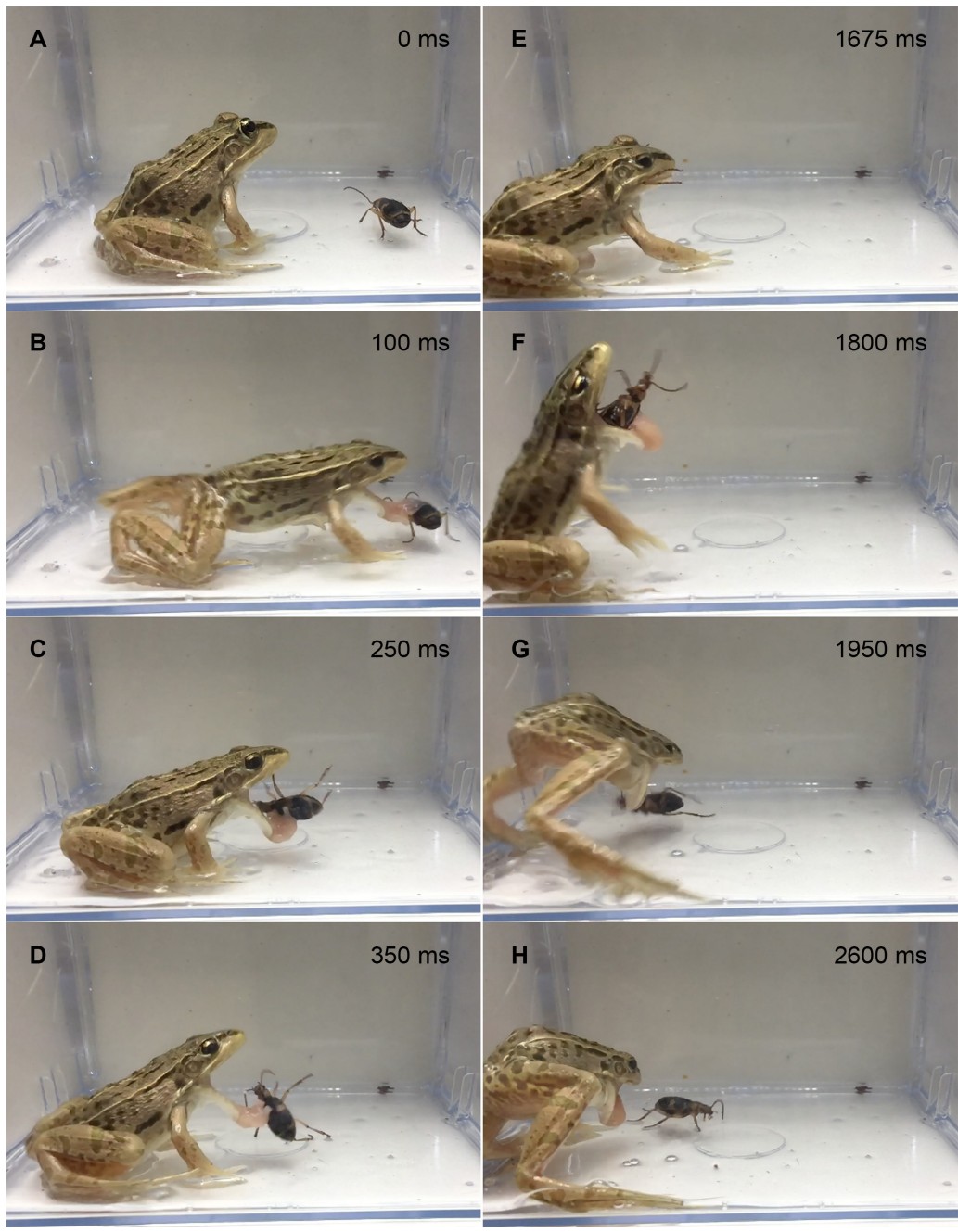

**Figure 3** **Temporal sequence of the frog *Pelophylax nigromaculatus* spitting out a live adult *Pheropso-phus jessoensis* after taking the beetle into its mouth.** (A) 0 ms. (B) 100 ms. (C) 250 ms. (D) 350 ms. (E) 1,675 ms. (F) 1,800 ms. (G) 1,950 ms (H) 2,600 ms. Bombing by the beetle was audible just before the frog spat out the beetle (1,675–1,800 ms; see Video S4). Credit: Shinji Sugiura.

**Table 1** Results of a generalised linear model (GLM) testing potential factors influencing whether the frog *Pelophylax nigromaculatus* successfully swallowed the bombardier beetle *Pheropsophus jessoensis* in feeding experiments.

| Response variable | Explanatory variable (fixed effect) | Coefficient estimate | SE | *t* value | *P* value |
|---|---|---|---|---|---|
| Swallowing success[a] | Intercept | 7.601881 | 4.454207 | 1.707 | 0.094 |
| | Frog weight | −0.420817 | 0.290774 | −1.447 | 0.154 |
| | Beetle weight | −0.060781 | 0.025084 | −2.423 | 0.019 |
| | Frog weight × Beetle weight | 0.002637 | 0.001457 | 1.81 | 0.076[c] |
| | Beetle treatment[b] | 1.387536 | 0.855616 | 1.622 | 0.111 |

Notes.

[a] As the residual deviance was smaller than the residual degrees of freedom, a quasi-binomial error distribution (rather than a binomial error distribution) was employed. Two and four live and dead beetles, respectively, were swallowed.

[b] Live beetles were used as a reference.

[c] The significance of this factor was checked using the likelihood ratio test ($P = 0.063$).

## DISUSSION

Here, I found that *Pe. nigromaculatus* frequently rejected *Ph. jessoensis* without attempting to swallow the beetles (Fig. 1). Around 70% of frogs stopped attacking both live and dead beetles immediately after touching the beetles with their tongues. Although the high-speed release of hot noxious chemicals (bombing) protected *Ph. jessoensis* from digestion by the frog *Pe. nigromaculatus* (Figs. 1 and 3; Video S4), my findings support the hypothesis that bombing is not essential for *Ph. jessoensis* to successfully evade being swallowed by *Pe. nigromaculatus* (Figs. 1 and 3; Video S3). Which factors, then, stopped the frogs from attacking? Three potential reasons can be considered: (1) the frogs recognised the warning colouration of the beetles; (2) the body size of the beetles was too large for the frog to accommodate; and (3) the frogs reflexively avoided the beetles after detecting toxic substances or other deterrent characteristics on the beetles' body surfaces.

The bombardier beetle *Ph. jessoensis* does have a striking yellow and black pattern on its body that could serve as warning colouration (Fig. 1), although this has not been empirically demonstrated. Anuran predators can avoid toxic prey by recognising certain colours or other morphological characteristics and then ignoring those prey (*Brower, Brower & Westcott, 1960*; *Brower & Brower, 1962*; *Dean, 1980a*; *Taniguchi et al., 2005*; *Ito, Taniguchi & Billen, 2016*). In fact, 34.1 and 29.2% of frogs did not seek to attack live or dead beetles, respectively, before the feeding experiments commenced, suggesting that *Pe. nigromaculatus* may recognise the body pattern and shape of *Ph. jessoensis* and interpret these as warning signals. However, frogs that ignored *Ph. jessoensis* did not respond to other prey (i.e., *T. molitor* larvae). Therefore, the experimental conditions used may not be appropriate for analysing foraging by certain frogs. Alternatively, the yellow-and-black pattern of *Ph. jessoensis* may serve as disruptive camouflage; flightless *Ph. jessoensis* walk on soil of forest edges, grasslands, and agricultural fields. Further work is needed to explore the significance of *Ph. jessoensis* colour as a defensive trait.

GLM analysis indicated that beetle size was correlated with beetle-swallowing frequency of frogs (Table 1). *Pelophylax nigromaculatus* has been reported to spit out large prey that they were unable to swallow after taking the prey into their mouths (*Honma, 2004*; *Honma,*
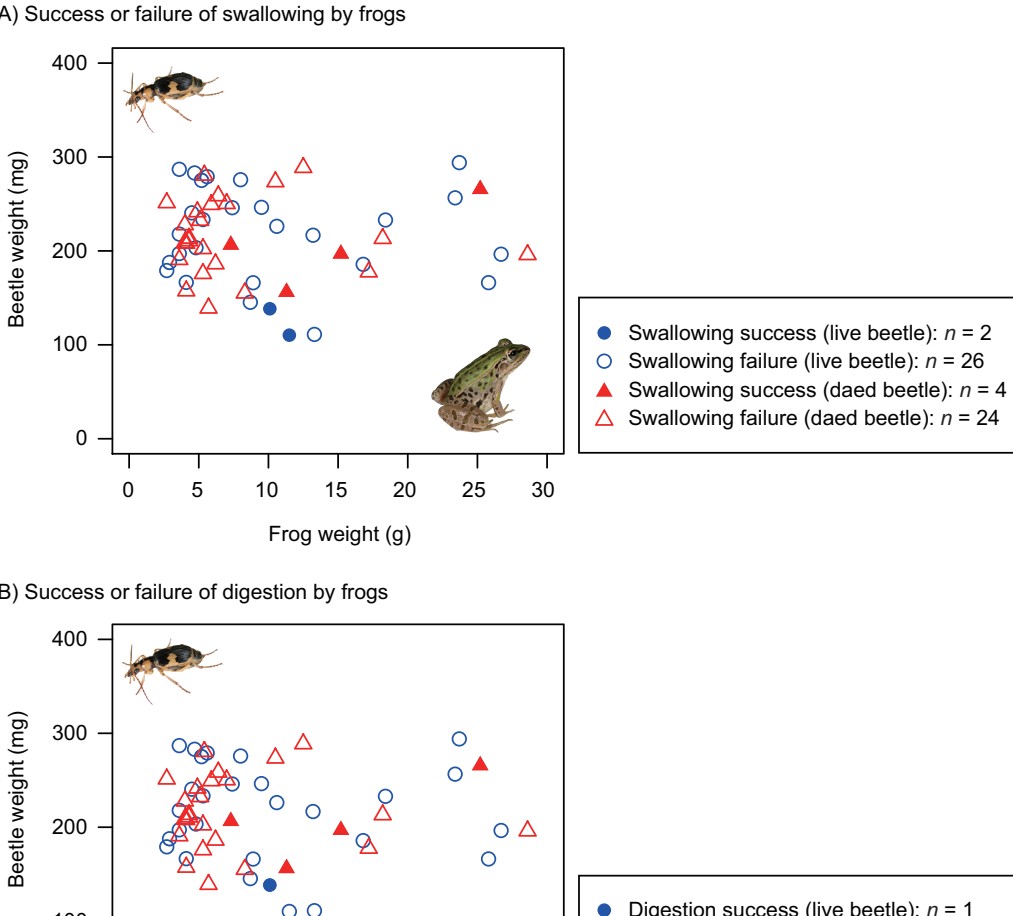

A) Success or failure of swallowing by frogs

- ● Swallowing success (live beetle): *n* = 2
- ○ Swallowing failure (live beetle): *n* = 26
- ▲ Swallowing success (daed beetle): *n* = 4
- △ Swallowing failure (daed beetle): *n* = 24

B) Success or failure of digestion by frogs

- ● Digestion success (live beetle): *n* = 1
- ○ Digestion failure (live beetle): *n* = 27
- ▲ Digestion success (daed beetle): *n* = 4
- △ Digestion failure (daed beetle): *n* = 24

**Figure 4** **Body size relationships between predator frogs (*Pelophylax nigromaculatus*) and prey beetles (*Pheropsophus jessoensis*).** (A) Success or failure of swallowing by frogs. (B) Success or failure of digestion by frogs. Closed circles and triangles indicate the swallow (or digestion) success of live and dead beetles, respectively. Open circles and triangles indicate the swallow (or digestion) failure of live and dead beetles, respectively. Photo credit: Shinji Sugiura.

*Oku & Nishida, 2006*). However, 67.9% of the frogs in the experiment with live beetles and 71.4% of the frogs in the experiment with dead beetles stopped their predatory attacks before taking the beetles into their mouths (Figs. 1 and 2; Video S3, Video S5). Thus, my results do not provide strong evidence that the frogs could not ingest large beetles. Rather, the amount of chemicals on the beetle body surface may increase with beetle size (see below).

**Table 2** Results of a generalised linear model (GLM) testing potential factors influencing whether the frog *Pelophylax nigromaculatus* successfully digested the bombardier beetle *Pheropsophus jessoensis* in feeding experiments.

| Response variable | Explanatory variable (fixed effect) | Coefficient estimate | SE | *t* value | *P* value |
|---|---|---|---|---|---|
| Digestion success[a] | Intercept | 4.471889 | 4.45551 | 1.004 | 0.32 |
| | Frog weight | −0.370908 | 0.30261 | −1.226 | 0.226 |
| | Beetle weight | −0.04681 | 0.023848 | −1.963 | 0.055[c] |
| | Frog weight × Beetle weight | 0.002329 | 0.00146 | 1.596 | 0.117 |
| | Beetle treatment[b] | 2.011385 | 1.033049 | 1.947 | 0.057[d] |

Notes.

[a] As the residual deviance was smaller than the residual degrees of freedom, a quasi-binomial error distribution (rather than a binomial error distribution) was employed. One and four live and dead beetles, respectively, were digested.

[b] Live beetles were used as a reference.

[c] The significance of this factor was checked using the likelihood ratio test ($P = 0.028$).

[d] The significance of this factor was checked using the likelihood ratio test ($P = 0.029$).

The rapid responses of the frog species *Pe. nigromaculatus* to bombardier beetles (Fig. 2) could be considered a reflex action of the frogs' tongues (cf. *Kumai, 1981a*; *Kumai, 1981b*; *Hirakawa, 1989*). Frogs are known to use their tongues as a chemical detector (*Dean, 1980b*; *Kumai, 1981a*; *Kumai, 1981b*; *Barlow, 1998*) as well as a prey-catching tool (*Noel et al., 2017*). For example, chemical or electrical stimulation of the tongue can generate reflex responses in *Pe. nigromaculatus* (*Kusano & Sato, 1957*; *Kumai, 1981a*; *Kumai, 1981b*; *Suzuki & Nomura, 1985*; *Takeuchi, Satou & Ueda, 1986*; *Hirakawa, 1989*). Because *Pe. nigromaculatus* is a generalist predator that can attack a variety of arthropods within its field of view (*Hirai & Matsui, 1999*; *Honma, 2004*; *Honma, Oku & Nishida, 2006*; *Sano & Shinohara, 2012*; *Sarashina, Yoshihisa & Yoshida, 2011*), *Pe. nigromaculatus* may have evolved specific responses to toxic prey to avoid being injured by trying to eat them. The results of this study suggest that the tongues of *Pe. nigromaculatus* may be able to rapidly detect toxic substances or other characteristics on the body surface of the bombardier beetles, and the frogs subsequently avoid the beetles to prevent themselves from being bombed and injured. Previous studies have focused on how frogs and toads use their tongues to catch prey (*Ewert, 1970*; *Nishikawa & Gans, 1996*; *Monroy & Nishikawa, 2010*; *Noel et al., 2017*). Few reports have explored how frogs and toads use their tongues to detect toxins in/on potential prey (but see *Dean, 1980b*). Therefore, the tongue responses that I describe in *Pe. nigromaculatus* will likely be evident in other frogs such as the tree frog *Hyla japonica* (Günther) (*Taniguchi et al., 2005*; *Ito, Taniguchi & Billen, 2016*; *Matsubara & Sugiura, 2017*).

## CONCLUSIONS

In one study, the chemicals produced by bombardier beetles' bombing did not stimulate the tongues of toads any less intensely than did the heat from the chemical reaction (*Dean, 1980b*). Other than this study, the relative importance of the toxic chemicals and heat produced by bombing for the successful escape of bombardier beetles from predators has been largely unexplored. My results support the hypothesis that bombing is not essential when bombardier beetles defend themselves against frog attacks. Furthermore, my findings

suggest that (cool) toxic chemicals on the beetles' bodies alone may cause frogs to desist from an attack; thus, chemicals on the body may serve as a primary defence and bombing as a secondary defence. Successful defence by chemicals on the body would reduce the costs associated with bombing, suggesting that beetles may have evolved to use chemicals on the body surface as their primary defence. However, further experiments are required to validate this hypothesis; for example, dead beetles with body surfaces cleaned of chemicals, or palatable prey coated with toxic chemicals, should be offered to frogs.

Many prey animals exhibit multiple anti-predator defences (*Edmunds, 1974*). Predation pressures imposed by different enemies may encourage prey to diversify defence strategies. Further studies are needed.

## ACKNOWLEDGEMENTS

I am grateful to K. Sakagami, K. Uchida, and A. Ushimaru for providing valuable information about the collection sites. I thank K. Sakagami, W. Higashikawa, and M. Ito for helping to collect the study animals, and S. Matsubara and Y. Maeda for helping to maintain them. T. Takanashi provided valuable advice on this research. M. Bulbert and two anonymous reviewers provided helpful comments on an earlier version of the manuscript.

### Funding

This study was financially supported by the Fujiwara Natural History Foundation (H28-23) and the Graduate School of Agriculture, Kobe University. The funders had no role in study design, data collection and analysis, decision to publish, or preparation of the manuscript.

### Grant Disclosures

The following grant information was disclosed by the author:
Fujiwara Natural History Foundation: H28-23.
Graduate School of Agriculture, Kobe University.

### Competing Interests

The authors declare there are no competing interests.

### Author Contributions

- Shinji Sugiura conceived and designed the experiments, performed the experiments, analyzed the data, contributed reagents/materials/analysis tools, prepared figures and/or tables, authored or reviewed drafts of the paper, approved the final draft.

### Animal Ethics

The following information was supplied relating to ethical approvals (i.e., approving body and any reference numbers):

The experiments were undertaken in accordance with the Kobe University Animal Experimentation Regulations (Kobe University's Animal Care and Use Committee, 27–01, 30–01).

### Data Availability
Figshare: https://doi.org/10.6084/m9.figshare.7087127.v1.

### Supplemental Information
Supplemental information for this article can be found online at http://dx.doi.org/10.7717/peerj.5942#supplemental-information.

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
