# Peer review of "Anti-predator defences of a bombardier beetle: is bombing essential for successful escape from frogs?"

_PeerJ, doi:10.7717/peerj.5942_

## Round 0.1 · original submission · Major Revisions

This is a very interesting manuscript, but the reviewers have a number of concerns which you must address. Please revise your paper in accordance with their comments. Be advised that the revised version will probably be returned to the original reviewers.

·

Basic reporting

Overall:
This paper entitled: "Anti-predator defences of a bombardier beetle: is bombing essential for successful escape from frogs?" attempts to determine, in a sense, that spraying is a secondary defensive strategy, and infers a primary defence is some recognition chemical on the surface of the beetles. This chemical is used by the frogs to detect if the beetle is acceptable to eat or not. I very much enjoyed this topic and I can see much potential for the topic area. However I feel the paper has a number of issues that need to be addressed before it can be accepted. I provide comments addressing specific reviewing criteria for PeerJ here. Further amendments and issues are provided on the PDF itself best viewed using acrobat.

A. Clear and unambiguous, professional English used throughout:
The article was well written and read well. So thanks to the reviewer for putting in the effort to have it checked.

B. Literature references, sufficient field background/context provided:
The introduction had some good content. I felt the discussion is missing a line or two on the presumed cost of using sprays. As such it misses the opportunity to better support a hypotheses that may not be obvious to the readers (i.e. bombardier beetles spray when grabbed so many might expect the beetles use this as a primary defence). If spraying is costly then you might expect that an animal that uses sprays may have a primary defence that is not using the spray. This has not been investigated in a predator-prey context.

C. Professional article structure, figs, tables. Raw data shared.
Figure 1 seems reasonable; Figure 2 and 3 are indeed cool but not necessary as the supplementary material has video which has the same info but in action. The captions of the tables need more information such as sample numbers, units, the z statistic used to calculate the P value (i.e. I'm assuming it's a Wald Z) and the P value cut off. Raw data has been shared as requested in figshare.

D. Self-contained with relevant results to hypotheses
Yes it adheres

Experimental design

A. Original primary research within Aims and Scope of the journal.
Yep

B. Research question well defined, relevant & meaningful. It is stated how research fills an identified knowledge gap
The research question is clear and straight forward. I felt it could have been a slightly bigger question though by thinking a bit more about the costs related to spraying to put the paper in a more generalised theoretical context so it is tackling a question that is applicable to all sprayers and not just bombardier beetles. Also I think a bit more on the concept that prey have both primary and secondary defences. Despite that the article is relatively clear.

C. Rigorous investigation performed to a high technical & ethical standard.
Fairly straight-forward experiment design with straight-forward data collection via video. For the most part this was done well and the videos were fabulous.

D. Methods described with sufficient detail & information to replicate.
Yes but some sections I feel would be better placed in the introduction rather than being results. On another note: I felt the paper missed an opportunity by not including the frogs that did not attack. I feel they should have been part of the experimental design. It would have been ideal to let them interact and if they did not attack after a certain time then offer them a carabidae they would eat. If they attack it then you would know if the frogs are visually recognising the bombardier and choosing not to attack, if they do not attack actual food then you would know for sure if they are simply not hungry. This would have made the text in the discussion about visual warning signals relevant.

Validity of the findings

A. Data is robust, statistically sound, & controlled.
a) The description of the results suggests there is a relationship between the size of the frog and the size of the beetle and the likelihood of the beetle being eaten. This intuitively makes sense but the stats do not back this up. The current coding of the variables requires an interaction to ascertain if this exists. Otherwise it could simply be smaller beetles are eaten irrespective of frog size. And big frogs are more likely to eat a beetle irrespective of the beetle size. If an interaction cannot be fitted due to insufficient data then some plot looking at the relationship between the two is required. An alternative approach to modelling this could be to just calculate the difference in size between frog and beetle and so it is then just one variable that indicates the difference in size so an interaction is not required. A good approach if the power from the number of replicates is not enough. Note though this provides an indication of the relative difference. If there is a reason for a frog of a certain size should not eat the beetles then your currently modelling procedure will provide that information with the interaction providing the relative component.
b) Need to state Wald Z statistics were used to estimate P-value. Given the marginal significance it is pertinent to use a log-likelihood ratio test to gain an estimate of the significance of each variable to the model. This is slightly more conservative and more reliable than Wald Z. This is just to check the marginal effects are indeed real or it's a type error caused by the method of estimation. If you are unaware of how to do this just google it but it involves comparing the significance of the model with and without the variable of interest.
c) The trend in figure 1 is not stated: I realise the stats are not significant but there is a definite trend in the figure that is interesting. Proportionally more live beetles were spat out and less were swallowed. This may imply a difference in the virulence of the 'chemical surface' or chemical bombing.
d) How the author has used the reference to underdispersion is a bit odd. Firstly, for binomial models the dispersion parameter is forced to be 1. However if indeed there is an issue with underdispersal this means it is harder to obtained a significant result i.e. your standard errors maybe wider than is actual the case. If this is a concern then the model has to modelled using something other than binomial so something like a quasi-binomial. In the end though the stating of the dispersal parameters mean nothing biologically they instead inform the type of modelling structure to best estimate parameters.
e) Having looked at the raw data it occurred to me that the frog and beetles are expressed in different units. Were the models specified using grams for both frog and beetle or the different units? This can influence the results as there is a big magnitude difference and also may contribute to the dispersal issues
f) An alternative approach to modelling this data could be to use a hazard model or even an ordinal model. Both approaches you would not then need to have two models one for grabbing and the beetles and one for swallowing. This is not a massive issue but from a theoretical point of view modelling the data as separate does not reflect reality as swallowing the carabidae cannot happen with first putting it in its mouth.

B. Conclusion are well stated, linked to original research question & limited to supporting results.
Discussion: is underdone:
1. Parts of the discussion were results and should be placed there.
2. It does not make much of the frogs responses to the dead bombardiers. A greater proportion of them were eaten. I realise it might not be significant but the trend is there. This is quite interesting as it does suggest the spray is important for preventing consumption and hence the use of it escalates with increased threat of digestion. Likewise it could be due to the 'detection chemical' breaking down after death and this has been the case for other toxic or distasteful products.
3. In terms of body size well it has been shown recently that with an increase in the size of the predator relative to the prey, the prey has to maintain a higher level of 'distasteful agent' so animals of larger size may have a greater capacity to withstand distasteful substances.
4. The use of the tongue by frogs to taste things and thus reject is known but I feel the discussion does not provide references that discuss this satisfactorily.
5. Again no reference to why the prey would use some form of surface recognition effective against frogs and waits to use the spray at a particular point of the subjugation stage. Some good points to raise there.

Additional comments

As stated above more comments, questions and recommendations are found on the PDF itself and can be viewed using acrobat or preview.

Reviewer 2 ·

Basic reporting

no comment

Experimental design

no comment

Validity of the findings

no comment

Additional comments

This study presents a simple experiment in which frogs were presented with dead or live bombardier beetles. The results indicate that dead insects are rejected at the same frequency as live ones, suggesting that deterrents other than active bombing exist.
The paper was clearly written and the experimental design and analyses generally adequate and appropriate. The still images from the video trials are a very useful indicator of how the behaviours were scored.

The results essentially comprise a single comparison of live vs dead beetles. The separate analyses of swallowing vs digesting success appear to differ by only a single observation. Potential reasons why dead beetles are rejected are discussed but further manipulations to clarify the mechanism (e.g. cleaning the surface of dead beetles, coating palatable prey with bombing chemicals) were not performed. Perhaps these and other manipulations should be mentioned as fruitful directions for future study.

Nor control trials were conducted to document rejection rates of non-noxious prey. Implicit in the interpretation of the study is that 'normal' prey are rarely rejected or spit out. Perhaps these frogs regularly refuse or spit out any prey under artificial conditions. Although it seems unlikely, it is conceivable that neither live nor dead bombardier beetles are rejected any more frequently than any other prey under these conditions. This possibility at least needs to be addressed, preferably with data.

Some specific comments-

L124. How long were frogs in captivity before they were used in feeding trials?
Were all frogs eating normally in captivity before trials? It would be useful to include data on rejection rates of non-bombardier beetle prey.

L159. Does 'contact' exclusively refer to the frogs touching the beetle with their tongue? Or did they sometimes touch it with another part of their body?

L196. How many frogs successfully digested dead beetles? 3?

L247. It would be useful to include the number of frogs that did not try to attack the beetle when it was outside the cage. This would give an indication of how many may have been deterred by visual cues

I suggest adding information on the total number of successes into the headings of Tables 1 (4 out of 46?) and 2 ( 3 out of 46?). Is the only difference between the analyses presented in Table 1 and 2 the status of a single observation (i.e. the one individual that vomited up a beetle after swallowing it)?

Reviewer 3 ·

Basic reporting

- The English is clear throughout
- background should be broadened to include information on similar species that use these kinds of defenses
- hypothesis/es not clearly articulated

Experimental design

Difficult to judge the appropriateness of the experimental design because the hypothesis is unclear.

Validity of the findings

results seem valid, but the hypothesis being tested is unlcear and obscured.

Additional comments

This data set is very valuable to the field of predator prey interactions, but I find the manuscript difficult to interpret. I think it’s primarily because the hypothesis is not articulated clearly and as such the aim of the experiment is obscured and it is difficult to tell whether the methods are appropriate. The reactions of frogs to bombardier beetles is interesting as is the fact that the beetles seem to be defended by previously sprayed chemical defenses, and also that the beetles were not induced to spray despite prior to the frog attacks. Unfortunately, it is unclear exactly what has been tested in this experiment. I strongly encourage rewriting of the ms for the eventual publication of these data. The author(s) might be interested in framing the frog behaviour in the predation sequence framework described by Endler (1991) Interactions between predators and prey in Behavioural Ecology: An Evolutionary Approach. P. 169-202. The emerging literature on deimatic displays may also be of interest.

Abstract

Information about the attacks in which the beetles sprayed the frogs is needed here.

Introduction

More background on the occurrence of this type of defence in nature would be of interest to broaden the introduction.

68-69: could also suggest that frogs and toads do not attempt to eat bombardier beetles.

84: the hypothesis being tested is not clear and needs to be expressed precisely.

Methods

101: provide a sample size, the number of beetles collected.

115: provide a sample size, the number of frogs collected.

137: explain what constituted “attacking behaviour”

142: provide how long you observed the frogs to see whether they vomited

134 – 144: great! Please provide earlier in the methods.

156-161: clarify whether this part of the method refers to both experiments (live and dead beetles?) or just the experiment with the dead beetles

184: clarify if the 65.4% is of the 92.3% that rejected or out of the total 26 frogs

191-192: measurement of repeat attacks were not mentioned I the methods, clarify.

198-199: ambiguous, clarify: “continued with predatory behaviour soon afterwards”. Were frogs provided with multiple attempts to attack beetles? If so, this needs to be part of the experimental design and described in the methods.

Results

Whole numbers rather than percentages would be more transparent.

209-210: this is very interesting!

Discussion

229 – 230: the choice of the beetle to bomb or not is an important variable to include in the analyses depending on your hypothesis! Especially because you are using swallowing as a response measure.

257 – 258: large body size would be attractive to a predator!

278 – 279: ah, here is the hypothesis, bring it up to the end of the introduction. The in-the-mouth bombing provides an interaction that makes it important that you highlight whether it is the surface chemicals question you are answering.

Figure 1.
The proportion of beetles that bombed should be represented in this figure for all stages of predation.

---

## Round 0.2 · Minor Revisions

There are just a few more changes that are necessary, according to reviewer 2. These changes should be easy to make.

Reviewer 2 ·

Basic reporting

no comment

Experimental design

no comment

Validity of the findings

no comment

Additional comments

I found the clarity of the manuscript has improved considerably after incorporation of my and other reviewer comments. I have a few minor suggestions to further improve clarity.

L197. specify what time frame constitutes 'soon'. i.e. within 60min?
L287. Do frogs need to take food into their mouth to assess its size? Is it possible that they could assess prey size using just the tongue?
L290. 'please' is not necessary here.
L291. It was not clear to me what potential reflex you referring to here. The first sentence says 'contact' but the context of the rest of the paragraph sounds like the reflex to release the beetle, not to contact it.

---

## Round 0.3 · accepted · Accept

Thank you for your changes, and congratulations on a very interesting paper!

#